# Uptake of Skilled Maternal Healthcare in Ethiopia: A Positive Deviance Approach

**DOI:** 10.3390/ijerph17051712

**Published:** 2020-03-05

**Authors:** Seman K. Ousman, Jeanette H. Magnus, Johanne Sundby, Mekdes K. Gebremariam

**Affiliations:** 1St Paul’s Hospital Millennium Medical College (SPHMMC), Addis Ababa 22728/1000, Ethiopia; 2Faculty of Medicine, University of Oslo, 1078 Oslo, Norway; j.h.magnus@medisin.uio.no; 3Department of Global Community Health and Behavioral Sciences, Tulane School of Public Health and Tropical Medicine, New Orleans, LA 70112, USA; 4Institute of Health and Society, HELSAM, University of Oslo, N-0316 Oslo, Norway; johanne.sundby@medisin.uio.no; 5Department of Nutrition, University of Oslo, 0317 Oslo, Norway; mekdes.gebremariam@medisin.uio.no

**Keywords:** positive deviance, skilled maternal healthcare, multilevel approach, DHS, Ethiopia

## Abstract

Risk factor approaches are often used when implementing programs aimed at enforcing advantageous health care behaviors. A less frequently-used strategy is to identify and capitalize on those who, despite risk factors, exhibit positive behaviors. The aim of our study was to identify positive deviant (PD) mothers for the uptake of skilled maternal services and to explore their characteristics. Data for the study came from two waves of the Ethiopian Demographic and Health Surveys conducted in 2011 and in 2016. PD mothers were defined as those reporting no formal education but with adequate use of antenatal care (ANC) and/or institutional delivery services. Two-level multilevel regression analysis was used to analyze the data. Factors associated with PD for the use of ANC services were: partner’s education status, involvement in household decision making, exposure to media, and distance to the health facility. Factors associated with PD for health facility delivery were: partner’s education, woman’s employment status, ANC visit during index pregnancy, exposure to media, and perceived challenge to reach health facility. Rural-urban and time-related differences were also identified. The positive deviance approach provides a means for local policy makers and program managers to identify factors facilitating improved health behaviour and ultimately better health outcomes while acknowledging adverse risk profiles.

## 1. Introduction

Maternal healthcare service utilization is an important predictor of favorable maternal and child health outcomes. Improving maternal and newborn healthcare is a priority of the Sustainable Development Goals (SDGs) [1]. In Ethiopia, 27–33 mothers die every day due to birth-related complications, equivalent to a maternal mortality ratio (MMR) of 412 per 100,000 live births [2]. Despite a decline from 871 per 100 000 live births in 2000 [3], much work is needed in Ethiopia to reach the political MMR goal of less than 70 deaths per 100,000 live births by 2030 [1]. Evidence suggests that maternal deaths are reduced by widespread utilization of skilled maternal healthcare services [4,5,6]. There is currently an underutilization of all maternal healthcare services by a significant proportion of women [2,7,8,9], and strategies and programs aimed at improving use of maternal health services have been implemented [10,11]. Despite these efforts, a recent report states that in Ethiopia less than half of all pregnant women received skilled maternal healthcare services (antenatal, delivery, and postnatal care) by a trained health professional at home or at health facilities [12,13].

Previous studies focusing on all women of reproductive age have identified factors influencing uptake and utilization of skilled maternal healthcare at multiple levels. Individual and household level factors identified are maternal age and education, household wealth, women’s decision-making power, parity, the woman’s or husband’s occupation, and media exposure [14,15]. Community level factors are geographical accessibility, residing in a community where women have higher levels of health knowledge, decision-making autonomy, low community poverty rate, and availability of community media [16]. At the policy level, government health strategies aimed at improving utilization of skilled maternal services have influenced uptake and utilization of maternal healthcare services [17,18,19].

Prior research from different areas of public health also shows that in every community there are certain individuals or groups whose behaviors and strategies enable them to find better solutions to common challenges than their peers, despite access to the same limited resources. These individuals are considered positive deviants (PDs) [20]. Their strategies result in improved health outcomes, favorable health behaviors or increased adherence to policy advice, despite exhibiting risk factors indicating otherwise [21,22,23,24,25,26,27]. Research adopting a PD strategy when exploring factors influencing the use of maternal healthcare services is however still in its infancy [22], as existing literature in general focuses on all women in the reproductive age [14,15,16,17,18,19]. Identifying Ethiopian mothers with positive health behaviors despite an adverse risk profile and exploring their characteristics could help health policy implementation, enable program managers to optimize public health initiative’s performance, and ultimately improve population health. Against this background, the current study aims to identify women at a high risk for underutilization of skilled maternal healthcare services and to explore factors that characterize the PD women for maternal healthcare services uptake in Ethiopia.

## 2. Materials and Methods

### 2.1. Data Source

We used data from the two latest Ethiopian Demographic Health Surveys (EDHS), conducted by the Ethiopian Central Statistical Agency (CSA) and ORC Macro International, USA, between December 2010–June 2011, and January 2016–June 2016. The full details of the methods and procedures used in the data collection of each EDHS, are published elsewhere [2,3,28]. The current study includes weighted data from 7584 women collected from 641 enumeration areas (EAs) (clusters) in 2016 and 7908 women from 595 EA clusters in 2011 (additional Appendix A). The eligibility criteria were: being of reproductive age (15 to 49 years); reporting at least one birth during the five years preceding the actual survey (i.e., 2006–2011 and 2012–2016); and participating in one of the two surveys from any region in the country.

### 2.2. Measurement of Variables

#### 2.2.1. Outcome measurement

The analyses in the current study addressed two maternity healthcare binary outcomes: (1) antenatal care (ANC) use, categorized into four or more visits (≥4) and less than four visits (<4), in accordance with the 2002 WHO ANC model; and (2) place of delivery, either home birth or birth at a health facility.

#### 2.2.2. Potential Predictors

Individual level: Age at the last birth, the birth order, education level of the woman and her partner, employment status of the participant and her partner, empowerment (related to household decision making and whether the woman was involved or not in aspects related to- her own health care, large household purchases, visits to family or relatives), household wealth index (low and high household wealth as calculated by demographic health survey (DHS) algorithm), mass media (radio, TV) exposure (no exposure, exposed to either a radio or TV, exposed to both), relationship status (being in a polygynous union or not), breastfeeding status (Yes/No), and perceived distance to a health facility to get medical help (‘yes, big problem’; or, ‘not big problem’).

Contextual community level: place of residence (urban or rural), and if the region was classified as agrarian, pastoral, or a city.

### 2.3. Identification of Positive Deviants

We used Anderson’s behavioral model of health service use [29], to identify positive deviants and the factors associated with being a PD. We selected women with no formal education as a sub group with very low likelihood of skilled maternal healthcare utilization, as education was the strongest predictor of both outcomes ANC and utilization of skilled health care during delivery after adjusting for the other risk factors associated with skilled maternity care in this population [2,28]. PD mothers were mothers who reported no formal education, but had an adequate use of ANC visits and or institutional delivery services. Thereafter, the analyses compared the characteristics of the PD mothers to those of their counterparts. Due to significant variations by place of residence in the overall use of skilled maternal healthcare, analyses were stratified by place of residence.

### 2.4. Statistical methods

#### Modeling Binary Responses

We used a binary logistic multilevel regression model, as the data was clustered at the survey level. We adjusted for confounders, decided a priori from the literature as age while giving last birth and order of the last birth. Bivariate logistic regression was performed to estimate the crude odds ratios (COR) and 95% confidence intervals of facility delivery or not, and if she had at least four ANC visits or not. Variables significantly associated with the outcome variable in the univariate analysis were entered in the multiple multilevel logistic regression analysis.

The study uses several explanatory variables that might be correlated to each other (such as maternal age at last birth and birth order). Multi-collinearity was checked using variance inflation factors (VIF) and variables with VIF less than 10 were considered for the analysis. In addition, we computed an estimate of intra-cluster correlation coefficient (ICC), which described the amount of variability in the response variables attributable to differences between the clusters. We used the McKelvey & Zavoina Pseudo R^2^ to assess the fit of the model [30,31]. Since the data were obtained from surveys conducted at two different time points, interactions with time were performed to describe any changes in adequate ANC services and health facility delivery among PDs in 2011 compared to 2016.

Sampling weights were applied for the data when we computed the univariate analysis to manage the unequal probability of selection between the strata defined by geographical location and for non-responses. Descriptive statistics were used to describe the characteristics of mothers. Bivariate analyses were first conducted. We then fitted two separate random-effects multilevel logistic regression models, one for each outcome of interest (ANC, and delivery care) using only the variables that are significantly associated with each outcome in the bivariate model. The model parameter estimates were obtained in the statistical software StataSE 15 using the restricted maximum likelihood method (REML). The level of significance was set at 0.05.

### 2.5. Ethical Consideration

The study adhered to national and international ethical guidelines for biomedical research involving human subjects [32], including the Helsinki declaration. The study was reviewed and approved by the Regional Committee for Medical and Health Research Ethics (Code number: 2016/967/REK sør-øst A) and the Norwegian Centre for Research Data (Code number: 48407) at the University of Oslo. Our team also requested permission and access to the data from the CSA in Ethiopia and Inner City Fund (ICF) international by registering online on the website www.dhsprogram.com [33] and submitting the study protocol (See, additional Appendix A) by highlighting the objectives of the study as part of the online registration process. The ICF Macro Inc removed all information that could be used to identify the respondents; hence, anonymity of the data was maintained.

## 3. Results

### 3.1. Characteristics of Participants

A total of 15,492 women reported a live birth in the past five-years preceding the surveys, 6720 (85.0%) (rural), and 1188 (15.0%) (urban) in 2011 (with mean age of 29.1 (±6.9) years), and 6619 (87.3%) (rural), and 965 (12.7%) (urban) in the 2016 survey, (29.3 (±6.8) years). Overall, in terms of pregnancy characteristics, both the number of antenatal care visits and the proportion of health facility delivery were consistently higher among urban women compared to rural dwellers. Indeed, more than 63% of urban mothers made the minimum four antenatal visits versus 27% of the rural mothers, and 84% of urban mothers reported health facility delivery versus 24% of the rural mothers in 2016. Table 1 presents the background characteristics of the women included.

### 3.2. Factors Associated with Positive Deviant Behaviour for Better Maternal Health Outcomes

#### 3.2.1. Factors Associated with Positive Deviant Behaviour for Antenatal Care

In 2011, among women with no formal education, 542 out of 4863 (11.2%) rural women and, 122 out of 407 (30.0%) urban women received at least four ANC services. In 2016 the numbers were 1050 out of 4562 (23.0%) rural women and 117 out of 229 (51.1%) urban women. These were classified as positive deviants (PDs) for ANC utilization. Between 2011 and 2016, the number of PDs for the uptake of adequate ANC services increased significantly by threefold (Adjusted Odds Ratio (AOR) = 3.01, (95% CI:2.55–3.55)) in rural areas, and by nearly double (AOR = 1.98, (95% CI:1.29–3.03)) in urban areas (Table 3 and Figure 1). There was no association with the order of the last birth or the number of under five children of PDs for the uptake of adequate ANC in the univariate models, so this was excluded from the multilevel models.

In 2011, rural mothers who were PDs for ANC uptake were more likely to be employed (*p* < 0.01), and in 2016, more likely to have husbands who were employed (*p* < 0.05), than their counterparts. The same was not true for urban PD mothers. The analysis also revealed that as compared to their counterparts, the rural PD women had partners with primary or above education (*p* < 0.05), were more likely to be involved in one or more decision making of the household (*p* < 0.05), and less likely to perceive a high distance to a health facility (*p* < 0.05) in both survey periods. In 2016, rural PDs were more likely to breastfeed (*p* < 0.05) compared to non-deviants’ peers. Both rural and urban PDs were more likely to report exposure to media (*p* < 0.05) in all surveys compared to non-deviants. In 2011, urban PD mothers were more likely to be from the city dwellers (*p* < 0.01), than non-deviants. Table 2 and Table 3 present the results of the multilevel regression analyses showing the characteristics of the PDs and ANC utilization in 2011 and 2016, respectively.

#### 3.2.2. Factors Associated with Positive Deviant Behaviour for Health Facility Delivery Utilization

In 2011, among women with no formal education, 160 out of 4863 (3.3%) rural women and 115 out of 407 (28.3%) urban women used health facility delivery services. Similarly, in 2016, 806 out of 4562 (17.7%) rural women, and 153 out of 229 (66.8%) urban women reported health facility delivery, and were classified as PDs. Table 4 and Table 5 presents the results of the multilevel logistic regression analyses showing the characteristics of the PDs in 2011 and 2016, respectively. Between 2011 and 2016, the odds of uptake of health facility delivery among PD women increased significantly, nearly by fivefold (AOR = 4.92, (95% CI:3.92–6.18)) in rural areas and by more than threefold (AOR = 3.23, (95% CI:2.03–5.15)) in urban dwellers (Table 5 and Figure 1).

In 2011, rural mothers who were PDs for institutional health care delivery were more likely to have a job (*p* < 0.01) than non-deviant counterparts. Both rural and urban PD women were more likely to have partners with primary or above education (*p* < 0.05), and in both surveys the PDs reported at least one ANC visit during pregnancy (*p* < 0.01). Likewise, in 2016, both urban and rural PDs were more likely to report exposure to either radio or TV than no exposure (*p* < 0.05) and to have breastfeed (*p* < 0.05) compared to non-deviants. In 2016, the rural PDs were more likely to be involved in one decision than no decision making for the household (*p* < 0.05); the same was not true for urban PDs. Rural mothers with birth order two or more (*p* < 0.01), and who perceive distance as a challenge to reach health facility (*p* < 0.05) were less likely to be PDs in both surveys. Lastly, in 2011, both rural and urban PDs for institutional health care delivery were more likely to be living in the city than agrarian communities (*p* < 0.01); rural PDs were also more likely to be living in pastoralist than agrarian communities.

## 4. Discussion

The study investigated factors associated with PD, i.e., an adequate use of ANC services or institutional delivery services despite no formal education, and thus classifying as at high-risk for non-use. In all models, compared to their counterparts, the odds of utilization of skilled maternal healthcare services (adequate ANC or health facility delivery) among PD women increased significantly from 2011 to 2016; the increase was higher in rural areas than in urban areas and higher for health facility delivery compared to ANC utilization uptake. Our study identified factors at multiple levels associated with PD behaviour. The results of this study are in concert with earlier studies reporting that level of education is associated with utilization of maternal healthcare services in Ethiopia and elsewhere [9,34,35,36,37,38]. Mothers with no education are at a particularly high risk of low utilization of maternal healthcare services and are consequently a key target group for intervention. Factors associated with PD behaviour might inform such interventions. The current study found that the employment status of women, partners’ education status, access to media, the level of women’s self-reported empowerment, and perceived adversity of distance to a health facility were associated with uptake of skilled maternal healthcare, and in concert with prior studies using risk factor analysis [39,40,41,42,43,44,45,46,47,48].

The relation between rural women’s employment and skilled maternal healthcare services among rural PDs might be related to employment making healthcare affordable, which might increase maternal healthcare service utilization. Moreover, employed women might be more likely to receive information at work that promotes health-seeking behaviour [49,50]. Interestingly, our analyses suggested that, compared to their counterparts, rural PD women with employed partners were more likely to used ANC services, but not health facility delivery, in the 2016 survey. This is interesting, because a woman whose husband is employed is likely to have better financial resources, which could facilitate access to skilled delivery. A study in Bangladesh showed that employed partners of PD women may spend more time away from their households than unemployed husbands, which might influence decisions about place of delivery [22].

In our study, both rural and urban women who were PDs for the uptake of skilled care were more likely to have partners with at least primary education. These findings are in line with findings of studies from Ethiopia and elsewhere suggesting that husband’s education may lead to greater involvement in maternity care utilization, as it is men who generally have an upper hand in decision-making at the household level in a patriarchal society like Ethiopia [51,52,53,54,55,56]. Similarly, we demonstrated that all PDs for the uptake of skilled care were more likely to report exposure to media compared to non-deviants in both surveys, irrespective of their location. Previous studies have shown that exposure to mass media at the individual and community level positively influences the development of positive behaviour toward the utilization of maternal healthcare services [57,58,59,60]. The impact of mass media exposure was however less consistent for health facility delivery in the present study. The current findings suggest that as compared to their counterparts, rural PD women for the uptake of skilled care were involved in one or more decision making of the household. This finding is consistent with those of previous studies from Ethiopia and elsewhere suggesting that women’s involvement in decision making is associated with utilization of skilled care [49,61,62,63,64,65,66]. However, involvement in two or more decisions was not associated with PD for health facility delivery, which is a surprising finding that needs further exploration.

We also found that rural PD were less likely to perceived distance to the health facility as a challenge to receive care in both surveys. It can be argued that with limited transport availability, as well as the mountainous and rugged topography in rural parts of the country, women find accessing skilled care very challenging [8,15,35,40,43]. In 2016, both rural and urban PDs for health facility delivery were more likely to breastfeed than their peers. This might reflect personal characteristics of these PD women, who might have a higher self-efficacy towards adopting healthier behaviours, better perceptions and norms regarding infant-feeding [67,68]. Both rural and urban PDs for health facility delivery were significantly more likely to have attended at least one ANC visit during index pregnancy. The role of usage of ANC when enhancing health facility delivery has been reported previously [9,34,37,47,48].

The findings of this study highlight factors at the socio-economic, behavioural, and structural levels associated with the uptake of skilled maternal healthcare services in an otherwise high-risk population. These factors should be taken into consideration when aiming to improve maternal healthcare among high-risk populations. Our study indicated that women in disadvantaged circumstances can still achieve good outcomes amidst a host of contextual barriers that usually predict poor health outcomes. Positive deviants in this context are shown to be women who have husbands with primary education or above, who have access to information through mass media, and who are involved at least with one decision made in the household. By knowing these characteristics, multiple government sectors (i.e., education, information and culture among others, alongside ministry of health) can identify women who fit the description and thus develop and amplify action plans encouraging more positive deviant behaviours. Finally, further qualitative research is warranted to explore in-depth what other exceptional characteristics PD women possess in order to shed light on potential mechanisms behind some of the associations identified in this study.

### Study Strength and Limitations

The study has several methodological strengths, including the use of multilevel modelling. The study was based on two different time points, and the data was nationally representative. The sample sizes of the surveys were large, providing high statistical power. The PD strategy offers an alternative to other approaches used in exploring the uptake of skilled maternal healthcare. Nonetheless, the PD sample size for urban women was small; thus, a lack of power might impact the multilevel analysis. The cross-sectional analyses at the two time points limited inference about causality. 

## 5. Conclusions

In this study, we identified mothers with no education as a group at high risk of poor uptake of maternal healthcare services. The women’s employment status, their partners’ education, access to media, self-reported empowerment, and perceived distance to reach a health facility were associated with PD behaviour and utilization of skilled maternal healthcare services. Differences between rural and urban PD women were documented. Indeed, the positive deviance approach provides a means for local policy makers and program managers to identify factors facilitating improved health behaviour, and ultimately better health outcomes, despite an acknowledged adverse risk profile. Such strategy and knowledge could facilitate targeted efforts aimed at achieving global SDGs of reduction of maternal mortality in resource-poor populations.

## Figures and Tables

**Figure 1 ijerph-17-01712-f001:**
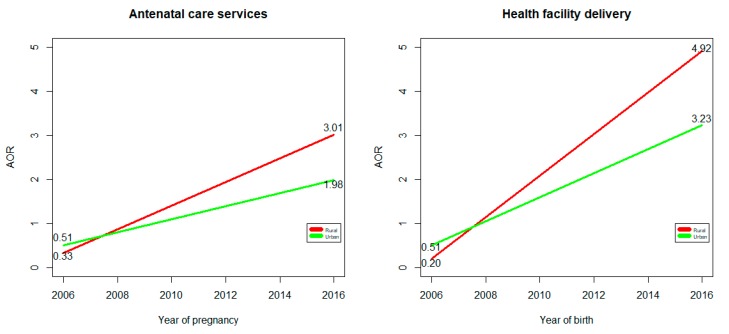
The changes in odds of uptake of skilled maternal healthcare service utilization (antenatal care or health facility delivery) among positive deviant women in urban and rural areas in Ethiopia between 2006 and 2016. Vertical axis = adjusted odds ratio (AOR); horizontal axis = survey periods.

**Table 1 ijerph-17-01712-t001:** Background characteristics of 15,492 women with a live birth in the five years preceding the 2011 or 2016 surveys (Ethiopian Demographic Health Survey (EDHS)).

EDHS Years	2011		2016	
Stratified by Place of Residence	Rural	Urban	Rural	Urban
Individual level Factors (N)	6720	1188	6619	965
Age of Mothers (Year) Mean (±SD)	29.3 (±7.1)	28.2 (±5.8)	29.4 (±7.1)	28.8 (±5.8)
Age when giving last birth (years)				
15—24	1691 (25.2)	319 (26.8)	1623 (24.5)	181 (18.7)
25—34	3234 (48.1)	638 (53.7)	3217 (48.6)	604 (62.6)
35—49	1795 (26.7)	231 (19.4)	1779 (26.9)	180 (18.7)
Order of the last birth				
First	1038 (15.4)	361 (30.4)	1100 (16.6)	331 (34.3)
Second or third	1990 (29.6)	472 (39.7)	1883 (28.4)	399 (41.3)
Fourth or higher	3692 (54.9)	355 (29.9)	3636 (54.9)	235 (24.4)
Marital status				
Not living with partner	567 (8.4)	156 (13.1)	393 (5.9)	87 (9.0)
Living with partner	6153 (91.6)	1032 (86.9)	6226 (94.1)	878 (91.0)
Education level of the women				
No education	4863 (72.4)	407 (34.3)	4562 (68.9)	229 (23.7)
Primary	1764 (26.2)	506 (42.6)	1833 (27.7)	315 (32.6)
Secondary and above	93 (1.4)	275 (23.1)	224 (3.4)	421 (43.6)
^b^ Working status of the women				
Working	2219 (33.0)	550 (46.3)	1702 (25.7)	466 (48.3)
Not working	4494 (66.9)	638 (53.7)	4917 (74.3)	499 (51.7)
Education level of their partners				
No education	3628 (54.0)	230 (19.3)	3200 (48.3)	146 (15.1)
Primary	2676 (39.8)	507 (42.7)	2480 (37.5)	251 (26.0)
Secondary and above	416 (6.2)	451 (38.0)	939 (14.2)	568 (58.9)
Working status of partners				
Working	6687 (99.5)	1172 (98.7)	6129 (92.6)	927 (96.1)
Not working	33 (0.5)	16 (1.3)	490 (7.4)	38 (3.9)
In a polygamous relationship				
Yes	1290 (19.2)	233 (19.6)	1094 (16.5)	148 (15.4)
No	5430 (80.8)	955 (80.4)	5525 (83.5)	817 (84.6)
^b^ Self-reported empowerment of women				
Not involved at all in decision making	795 (12.9)	60 (5.8)	764 (12.3)	25 (2.8)
Involved in one	891(14.5)	90 (8.7)	457 (7.3)	33 (3.8)
Involved in two	1373 (22.3)	189 (18.3)	793 (12.7)	122 (13.9)
Involved in at least three	3093 (50.3)	694 (67.2)	4212 (67.6)	698 (79.5)
Household wealth index				
Low household wealth status	5015 (74.6)	48 (4.1)	4815 (72.7)	79 (8.1)
High household Wealth status	1705 (25.4)	1140 (95.9)	1804 (27.3)	886 (91.9)
Exposure to mass media				
No exposure	3051 (45.4)	163 (13.7)	4831 (73.0)	192 (19.9)
Exposed to either radio or TV	2246 (33.4)	346 (29.1)	1207 (18.2)	335 (34.7)
Exposed to both radio and TV	1423 (21.2)	679 (57.2)	581 (8.8)	438 (45.4)
^b^ Ever had a terminated pregnancy				
Yes	740 (11.0)	96 (8.1)	603 (9.1)	77 (8.0)
No	5980 (89.0)	1092 (91.9)	6016 (90.9)	888 (92.0)
Number of under 5 children				
No child	278 (4.1)	91 (7.7)	219 (3.3)	63 (6.5)
One child	2844 (42.3)	720 (60.6)	3101 (46.9)	638 (66.1)
Two or more	3598 (53.5)	377 (31.7)	3299 (49.8)	264 (27.4)
Currently breastfeeding				
Yes	4573 (68.1)	651 (54.8)	4277 (64.6)	534 (55.4)
No	2147 (31.9)	537 (45.2)	2342 (35.4)	431 (44.6)
^b^ Perceived distance to a health facility to get medical help				
Big problem	5360 (79.8)	397 (33.6)	4244 (64.1)	162 (16.8)
Not a big problem	1356 (20.2)	786 (66.4)	2375 (35.9)	803 (83.2)
Anemia status during pregnancy				
Anemic	1472 (21.9)	235 (19.8)	2076 (31.4)	224 (23.2)
Not Anemic	5248 (78.1)	953 (80.2)	4543 (68.6)	741 (76.8)
Community level factors				
Contextual Regions				
Agrarian	6406 (95.3)	865 (72.8)	6223 (94.0)	671 (69.6)
Pastoralist	291 (4.3)	108 (9.1)	370 (5.6)	71 (7.3)
City dweller’s	23 (0.3)	215 (18.1)	26 (0.4)	223 (23.1)
Maternal Healthcare Service				
ANC Utilization				
No ANC visits	4249 (63.2)	295 (24.8)	2734 (41.3)	99 (10.3)
At least one ANC visit	2471 (36.8)	893 (75.2)	3885 (58.7)	866 (89.7)
At least 4 antenatal care visits	973 (14.5)	562 (47.3)	1815 (27.4)	609 (63.1)
Below 4 antenatal care visits	5747 (85.5)	626 (52.7)	4804 (72.6)	356 (36.9)
Place of delivery				
Home birth	6389 (95.1)	558 (47.0)	5024 (75.9)	155 (16.1)
Health Institution	331 (4.9)	630 (53.0)	1595 (24.1)	810 (83.9)

^b^ Total figure may not add to 100 percent due to ‘do not know’ and ‘missing cases’.

**Table 2 ijerph-17-01712-t002:** Results of multiple multilevel logistic regression exploring factors associated with being a positive deviant for antenatal care utilization among urban and rural women, 2011 Ethiopia DHS.

Survey Years	2011 Model					
Stratified by (N)	Rural (4863)			Urban (407)		
Deviance Status	PDs N (%)	Non-PDs N (%)	AOR (95 % CI)	PDs N (%)	Non-PDs N (%)	AOR (95% CI)
Co-variates (n)	542 (11.2%)	4321 (88,8%)	AOR (95 % CI)	122 (30.0%)	285 (70.0%)	AOR (95% CI)
Age when giving last birth (years)						
15—24 (ref)	87 (8.9)	893 (91.1)	1 (1,1)	17 (22.9)	57 (77.1)	1 (1,1)
25—34	288 (11.7)	2171 (88.3)	1.26 (0.91, 1.73)	72 (33.5)	145 (66.5)	1.03 (0.50, 2.12)
35—49	167 (11.7)	1257 (88.3)	**1.42 (1.00, 2.01) ****	33 (28.4)	83 (71.6)	1.25 (0.52, 3.00)
^b^ Working status of the women						
Not-working (ref)	298 (8.9)	3038 (91.1)	1 (1,1)	68 (32.9)	140 (67.1)	1 (1,1)
Working	243 (16.0)	1278 (84.0)	**1.64 (1.28, 2.10) *****	54 (27.2)	145 (72.8)	1.06 (0.60, 1.87)
Education level of their partners						
No education (ref)	302 (9.6)	2864 (90.4)	1 (1,1)	48 (28.5)	122 (71.5)	1 (1,1)
Primary	213 (13.6)	1353 (86.4)	**1.47 (1.15, 1.88) *****	55 (30.4)	126 (69.6)	1.17 (0.64, 2.11)
Secondary or above	27 (20.6)	104 (79.4)	**1.90 (1.10, 3.27) ****	19 (34.3)	37 (65.7)	1.26 (0.56, 2.88)
Working status of partners						
Not- working (ref)	6 (23.8)	19 (76.2)	1 (1,1)	9 (32.8)	5 (67.2)	1 (1,1)
Working	536 (11.1)	4302 (88.9)	1.38 (0.35, 5.46)	113 (32.8)	280 (67.2)	1.23 (0.33, 4.61)
^b^ Self-reported empowerment of women						
Not involved at all in decision making (ref)	34 (5.4)	599 (94.6)	1 (1,1)	5 (14.8)	28 (85.2)	1 (1,1)
Involved in one	63 (9.8)	582 (90.2)	1.25 (0.78, 1.99)	2 (4.4)	42 (95.6)	0.34 (0.10, 1.20)
Involved in two	123 (12.4)	868 (87.6)	**1.77 (1.16, 2.71) *****	22 (27.7)	56 (72.3)	0.58 (0.21, 1.61)
Involved in at least three	287 (13.0)	1917 (87.0)	**1.73 (1.16, 2.56) *****	78 (40.8)	113 (59.2)	1.22 (0.48, 3.12)
Household wealth index						
Low HH wealth status(ref)	376 (9.6)	3554 (90.4)	1 (1,1)	1 (3.9)	30 (96.1)	1 (1,1)
High HH wealth status	166 (17.8)	767 (82.2)	**1.77 (1.35, 2.32) *****	121 (32.3)	255 (67.7)	**3.65 (1.01, 13.13) ****
Exposure to mass media						
No exposure (ref)	177 (7.2)	2295 (92.8)	1 (1,1)	27 (24.1)	84 (75.9)	1 (1,1)
Exposed to either radio or TV	210 (13.3)	1363 (86.7)	1.26 (0.97, 1.63)	36 (24.5)	112 (75.5)	1.31 (0.65, 2.63)
Exposed to both radio and TV	155 (19.0)	663 (81.0)	**1.90 (1.38, 2.61) *****	59 (40.3)	89 (59.7)	**2.80 (1.32, 5.94) *****
Currently breastfeeding						
No (ref)	191 (12.7)	1313 (87.3)	1 (1,1)	52 (31.2)	117 (68.8)	1 (1,1)
Yes	351 (10.4)	3008 (89.6)	0.80 (0.64, 1.02)	70 (29.4)	168 (70.6)	0.74 (0.43, 1.27)
^b^ Perceived distance to a health facility to get medical help						
Not-big problem (ref)	135 (15.1)	762 (84.9)	1 (1,1)	80 (31.4)	177 (68.6)	1 (1,1)
Big problem	407 (10.3)	3555 (89.7)	**0.71 (0.54, 0.95) ****	42 (28.0)	108 (72.0)	1.05 (0.59, 1.87)
Anemia status						
Non-anemic(ref)	450 (12.0)	3300 (88.0)	1 (1,1)	104 (33.5)	208 (66.5)	1 (1,1)
Anemic	92 (8.2)	1021 (91.8)	0.81 (0.62, 1.06)	18 (18.9)	77 (81.1)	0.70 (0.38, 1.29)
Community level Factors						
Contextual Region						
Agrarian (ref)	525 (11.4)	4076 (88.6)	1 (1,1)	79 (26.8)	215 (73.2)	1 (1,1)
Pastoralist	15 (6.1)	228 (93.9)	**0.65 (0.44, 0.96) ****	8 (12.2)	54 (87.8)	0.72 (0.27, 1.88)
City dweller’s	2 (11.7)	17 (88.3)	0.87 (0.48, 1.59)	35 (71.0)	16 (29.0)	**4.92 (2.02, 12.00) *****

sig. at ** sig. at 5% level(in bold); *** sig. at 1% level (in bold); ^b^ Total figure may not add to 100 percent due to ‘do not know’ and ‘missing cases’. ref = reference group; PDs = Positive Deviants; N = Number of participants., AOR = Adjusted Odds ratios. Note: PD defined as illiterate women with an adequate use of antenatal care services.

**Table 3 ijerph-17-01712-t003:** Results of multiple multilevel logistic regression exploring factors associated with being a positive deviant for antenatal care utilization among urban and rural women, 2016 Ethiopia DHS.

Survey Years	2016 Model					
Stratified by (N)	Rural (4562)			Urban (229)		
Deviance Status	PDs N (%)	Non-PDs N (%)	AOR (95 % CI)	PDs N (%)	Non-PDs N (%)	AOR (95 % CI)
Co-variates (n)	1050 (23%)	3512 (77.0%)	AOR (95 % CI)	117 (51.1%)	112 (48.9%)	AOR (95 % CI)
Overall time effect (ref: 2011)			**3.01 (2.55, 3.55 ) *****			**1.98 (1.29, 3.03) *****
Age when giving last birth (years)						
15—24(ref)	119 (17.3)	567 (82.7)	1 (1,1)	13 (80.5)	3 (19.5)	1 (1,1)
25—34	608 (25.6)	1763 (74.4)	**1.35 (1.03, 1.77) ****	75 (51.3)	72 (48.7)	0.49 (0.21, 1.11)
35—49	323 (21.5)	1182 (78.5)	1.11 (0.83, 1.49)	29 (44.0)	37 (56.0)	**0.41 (0.17, 0.99) ****
^b^ Working status of the women						
Not-working(ref)	770 (22.4)	2664 (77.6)	1 (1,1)	60 (47.4)	66 (52.6)	1 (1,1)
Working	280 (24.8)	848 (75.2)	1.00 (0.80, 1.25)	57 (55.8)	45 (44.2)	1.08 (0.63, 1.87)
Education level of their partners						
No education(ref)	559 (20.7)	2145 (79.3)	1 (1,1)	43 (40.6)	62 (59.4)	1 (1,1)
Primary	377 (26.0)	1073 (74.0)	**1.32 (1.07, 1.62) ****	46 (64.4)	26 (35.6)	1.28 (0.71, 2.30)
Secondary or above	114 (27.9)	294 (72.1)	**1.63 (1.06, 2.48) ****	28 (54.2)	24 (45.8)	1.18 (0.58, 2.42)
Working status of partners						
Not-working(ref)	48 (13.0)	320 (87.0)	1 (1,1)	11 (61.9)	7 (38.1)	1 (1,1)
Working	1002 (23.9)	3192 (76.1)	**1.52 (1.07, 2.14) ****	106 (50.3)	105 (49.7)	1.07 (0.52, 2.18)
^b^ Self-reported empowerment of women						
Not involved at all in decision making(ref)	91 (16.2)	470 (83.8)	1 (1,1)	2 (20.0)	8 (80.0)	1 (1,1)
Involved in one	81 (25.9)	231 (74.1)	**1.67 (1.11, 2.52) ****	4 (69.5)	2 (30.5)	3.10 (0.72, 13.34)
Involved in two	145 (25.7)	419 (74.3)	1.17 (0.81, 1.69)	12 (37.5)	21 (62.5)	0.84 (0.23, 3.03)
Involved in at least three	657 (22.9)	2210 (77.1)	**1.43 (1.07, 1.92) ****	84 (54.5)	70 (45.5)	1.98 (0.67, 5.90)
Household wealth index						
Low HH wealth status(ref)	756 (21.3)	2801 (78.7)	1 (1,1)	15 (27.6)	38 (72.4)	1 (1,1)
High HH wealth status	294 (29.3)	711 (70.0)	1.20 (0.93, 1.56)	102 (58.2)	74 (41.8)	1.46 (0.71, 3.00)
Exposure to mass media						
No exposure (ref)	754 (21.2)	2807 (78.8)	1 (1,1)	45 (41.8)	63 (58.2)	1 (1,1)
Exposed to either radio or TV	207 (29.8)	487 (70.2)	**1.38 (1.07, 1.78)****	39 (52.3)	35 (47.7)	**1.88 (1.06, 3.35) ****
Exposed to both radio and TV	89 (28.9)	218 (71.1)	1.27 (0.85, 1.90)	33 (71.0)	14 (29.0)	**4.71 (2.10, 10.59)*****
Currently breastfeeding						
No (ref)	327 (19.4)	1353 (80.6)	1 (1,1)	41 (42.5)	55 (57.5)	1 (1,1)
Yes	723 (19.4)	2159 (80.6)	**1.24 (1.03, 1.50) ****	76 (57.4)	57 (42.6)	1.55 (0.93, 2.59)
^b^ Perceived distance to a health facility to get medical help						
Not-big problem(ref)	468 (29.5)	1120 (70.5)	1 (1,1)	96 (54.3)	81 (45.7)	1 (1,1)
Big problem	582 (19.6)	2392 (80.4)	**0.73 (0.60, 0.90) *****	21 (40.3)	31 (59.7)	0.94 (0.53, 1.67)
Anemia status						
Non-anemic(ref)	714 (23.5)	2322 (76.5)	1 (1,1)	93 (51.2)	89 (48.8)	1 (1,1)
Anemic	336 (22.0)	1190 (78.0)	0.98 (0.81, 1.20)	24 (51.1)	23 (48.9)	0.61 (0.35, 1.05)
Community level Factors						
Contextual Region						
Agrarian(ref)	1006 (23.7)	3237 (76.3)	1 (1,1)	85 (52.0)	79 (48.0)	1 (1,1)
Pastoralist	36 (11.9)	264 (88.1)	**0.49 (0.35, 0.69) *****	13 (33.1)	25 (66.9)	**0.44 (0.21, 0.92) ****
City dweller’s	8 (40.6)	11 (59.4)	1.27 (0.72, 2.24)	19 (71.6)	8 (28.4)	0.82 (0.39, 1.71)

sig. at ** sig. at 5% level (in bold); *** sig. at 1% level (in bold); ^b^ Total figure may not add to 100 percent due to ‘do not know’ and ‘missing cases’. ref = reference group; PDs = Positive Deviants; N = Number of participants., AOR = Adjusted Odds ratios. Note: PD defined as illiterate women with an adequate use of antenatal care services.

**Table 4 ijerph-17-01712-t004:** Results of multiple multilevel logistic regression exploring factors associated with being a positive deviant for having a health facility delivery among urban and rural women, 2011 Ethiopia DHS.

Survey Years	2011 Model					
Stratified by (N)	Rural (4863)			Urban (407)		
Deviant Status	PDs N (%)	Home Birth N (%)	AOR (95 % CI)	PDs N (%)	Home Birth N (%)	AOR (95 % CI)
Co-variates (n)	160 (3.3%)	4703 (96.7%)	AOR (95 % CI)	115 (28.3%)	292 (71.7%)	AOR (95 % CI)
Age when giving last birth (years)						
15—24(ref)	41 (4.2)	939 (95.8)	1 (1,1)	19 (25.9)	55 (74.1)	1 (1,1)
25—34	82 (3.3)	2378 (96.7)	1.46 (0.84, 2.54)	76 (34.9)	141 (65.1)	0.91 (0.38, 2.17)
35—49‘	37 (2.6)	1386 (97.4)	1.45 (0.74, 2.86)	20 (17.0)	96 (83.0)	0.67 (0.22, 2.03)
Order of the last birth						
First(ref)	37 (7.1)	494 (92.9)	1 (1,1)	21 (29.6)	49 (70.4)	1 (1,1)
Second or third	38 (2.7)	1344 (97.3)	**0.30 (0.16, 0.56) *****	49 (32.2)	103 (67.8)	0.60 (0.23, 1.58)
Fourth or higher	85 (2.9)	2865 (97.1)	**0.32 (0.16, 0.63) *****	45 (24.3)	140 (75.7)	0.60 (0.20, 1.79)
^b^ Working status of the women						
Not-working(ref)	77 (2.3)	3258 (97.7)	1 (1,1)	60 (28.7)	148 (71.3)	1 (1,1)
Working	82 (5.4)	1438 (94.6)	**1.77 (1.21, 2.59) *****	55 (27.6)	144 (72.4)	0.73 (0.39, 1.35)
Education level of their partners						
No education(ref)	102 (3.2)	3064 (96.8)	1 (1,1)	32 (20.3)	136 (79.7)	1 (1,1)
Primary	52 (3.3)	1513 (96.7)	**1.62 (1.12, 2.36) ****	58 (30.1)	124 (69.9)	1.49 (0.79, 2.82)
Secondary and above	6 (4.2)	126 (95.8)	1.55 (0.67, 3.61)	25 (60.0)	32 (40.0)	**3.05 (1.26, 7.36) ****
Working status of partners						
Not-working(ref)	3 (11.2)	22 (88.8)	1 (1,1)	3 (2.5)	13 (97.5)	1 (1,1)
Working	157 (3.2)	4681 (96.8)	0.34 (0.10, 1.20)	112 (29.1)	279 (70.9)	1.06 (0.24, 4.75)
^b^ Self-reported empowerment of women						
Not involved at all in decision making(ref)	13 (3.7)	620 (98.0)	1 (1,1)	7 (20.4)	27 (79.6)	1 (1,1)
Involved in one	16 (2.6)	629 (97.4)	**0.53 (0.29, 0.96) ****	13 (29.1)	31 (70.9)	0.90 (0.25, 3.30)
Involved in two	31 (3.1)	959 (96.9)	0.60 (0.34, 1.05)	12 (15.7)	66 (84.3)	1.20 (0.39, 3.72)
Involved in at least three	79 (3.6)	2125 (96.4)	0.67 (0.40, 1.11)	60 (31.7)	130 (68.3)	1.41 (0.47, 4.17)
Household wealth index						
Low HH wealth status(ref)	108 (2.7)	3822 (97.3)	1 (1,1)	2 (5.2)	29 (94.8)	1 (1,1)
High HH wealth status	52 (5.6)	881 (94.4)	1.26 (0.81, 1.95)	113 (30.1)	263 (69.9)	1.47 (0.46, 4.67)
Exposure to mass media						
No exposure (ref)	79 (3.2)	2393 (96.8)	1 (1,1)	17 (15.4)	93 (84.6)	1 (1,1)
Exposed to either radio or TV	52 (3.3)	1521 (96.7)	0.88 (0.59, 1.31)	43 (28.7)	106 (71.3)	1.01 (0.48, 2.11)
Exposed to both radio and TV	29 (3.5)	789 (96.5)	0.91 (0.53, 1.58)	55 (37.2)	93 (62.8)	1.62 (0.73, 3.60)
Number of under 5 children						
No child(ref)	18 (9.7)	161 (90.3)	1 (1,1)	7 (30.1)	16 (69.9)	1 (1,1)
One child	84 (4.4)	1847 (95.6)	**0.35 (0.17, 0.73) *****	57 (28.1)	145 (71.9)	3.52 (0.79, 15.65)
Two or more	58 (2.1)	2695 (97.9)	**0.25 (0.12, 0.53) *****	51 (28.0)	131 (72.0)	3.02 (0.64, 14.20)
Currently breastfeeding						
No(ref)	55 (3.6)	1449 (96.4)	1 (1,1)	42 (24.5)	127 (74.5)	1 (1,1)
Yes	105 (3.1)	3254 (96.9)	1.23 (0.83, 1.82)	73 (30.8)	165 (69.2)	1.35 (0.74, 2.46)
^b^ Perceived distance to a health facility to get medical help						
Not-big problem(ref)	51 (5.7)	845 (94.3)	1 (1,1)	73 (28.4)	184 (71.6)	1 (1,1)
Big problem	109 (2.7)	3854 (97.3)	**0.64 (0.42, 0.97) ****	42 (27.7)	108 (72.3)	0.65 (0.35, 1.19)
Anemia status						
Non-anemic(ref)	124 (3.3)	3627 (96.7)	1 (1,1)	97 (30.9)	215 (69.1)	1 (1,1)
Anemic	36 (3.2)	1076 (96.8)	0.89 (0.60, 1.31)	18 (19.3)	77 (80.7)	**0.41 (0.21, 0.79) *****
ANC Utilization						
No ANC visits(ref)	66 (2.0)	3258 (98.0)	1 (1,1)	21 (12.5)	146 (87.5)	1 (1,1)
At least one ANC visit	94 (6.1)	1445 (93.9)	**2.77 (1.90, 4.04) *****	94 (39.1)	146 (60.9)	**2.89 (1.54, 5.43) *****
Community level Factors						
Contextual Regions						
Agrarian(ref)	146 (3.2)	4456 (96.8)	1 (1,1)	71 (24.1)	223 (75.9)	1 (1,1)
Pastoralist	13 (5.3)	230 (94.7)	**2.53 (1.48, 4.32) *****	10 (16.1)	52 (83.9)	1.08 (0.39, 2.98)
City dweller’s	1 (6.3)	17 (93.7)	**2.78 (1.22, 6.32) ****	34 (66.2)	17 (33.8)	**10.12 (3.85, 26.61) *****

sig. at ** sig. at 5% level (in bold); ***sig. at 1% level (in bold); ^b^ Total figure may not add to 100 percent due to ‘do not know’ and ‘missing cases’. ref = reference group; PDs = positive deviants; N = number of participants., AOR = adjusted odds ratios. Note: PD defined as illiterate women with use of institutional delivery services.

**Table 5 ijerph-17-01712-t005:** Results of multiple multilevel logistic regression exploring factors associated with being a positive deviant for having a health facility delivery among urban and rural women, 2016 Ethiopia DHS.

Survey Years	2016 Model					
Stratified by (N)	Rural (4562)			Urban (229)		
Deviant Status	PDs N (%)	Home Birth N (%)	AOR (95 % CI	PDs N (%)	Home Birth N (%)	AOR (95 % CI)
Co-variates (n)	806 (17.7%)	3756 (82.3%)	AOR (95 % CI)	153 (66.8%)	76 (31.2%)	AOR (95 % CI)
Overall time effect (ref: 2011)			**4.92 (3.92, 6.18) *****			**3.23 (2.03, 5.15) *****
Age when giving last birth (years)						
15—24(ref)	135 (19.7)	551 (80.3)	1 (1,1)	10 (68.1)	6 (31.9)	1 (1,1)
25—34	402 (17.0)	1968 (83.0)	1.08 (0.73, 1.58)	108 (73.4)	39 (26.6)	1.16 (0.44, 3.01)
35—49	269 (17.8)	1237 (82.2)	1.37 (0.88, 2.14)	35 (53.2)	31 (46.8)	1.05 (0.35, 3.14)
Order of the last birth						
First(ref)	106 (28.2)	270 (71.8)	1 (1,1)	24 (84.3)	5 (15.7)	1 (1,1)
Second or third	216 (18.5)	950 (81.5)	**0.47 (0.30, 0.75) *****	51 (70.3)	22 (29.7)	0.71 (0.25, 2.05)
Fourth or higher	484 (16.0)	2536 (84.0)	**0.45 (0.27, 0.74) *****	78 (61.4)	49 (38.6)	0.55 (0.18, 1.70)
^b^ Working status of the women						
Not-working(ref)	575 (16.7)	2859 (83.3)	1 (1,1)	81 (64.5)	45 (35.5)	1 (1,1)
Working	231 (20.4)	897 (79.6)	1.14 (0.89, 1.47)	72 (70.5)	30 (29.5)	0.94 (0.50, 1.76)
Education level of their partners						
No education(ref)	425 (15.7)	2278 (84.3)	1 (1,1)	64 (61.9)	41 (38.1)	1 (1,1)
Primary	289 (19.9)	1162 (80.1)	**1.34 (1.05, 1.70) ****	50 (69.6)	22 (30.4)	0.96 (0.49, 1.87)
Secondary and above	92 (22.4)	316 (77.6)	**1.98 (1.25, 3.14) *****	39 (74.6)	13 (25.4)	1.73 (0.79, 3.79)
Working status of partners						
Not-working(ref)	49 (13.3)	319 (86.7)	1 (1,1)	11 (70.1)	6 (29.9)	1 (1,1)
Working	757 (18.0)	3437 (82.0)	1.30 (0.87, 1.93)	142 (67.0)	70 (33.0)	0.84 (0.38, 1.86)
^b^ Self-reported empowerment of women						
Not involved at all in decision making(ref)	83 (14.8)	478 (85.2)	1 (1,1)	5 (45.2)	6 (54.8)	1 (1,1)
Involved in one	65 (20.9)	246 (79.1)	**1.68 (1.04, 2.69) ****	4 (67.2)	2 (32.8)	1.10 (0.23, 5.19)
Involved in two	107 (18.9)	457 (81.1)	1.19 (0.78, 1.82)	14 (43.6)	19 (56.4)	1.38 (0.38, 5.02)
Involved in at least three	503 (17.5)	2364 (82.5)	1.14 (0.81, 1.61)	116 (74.6)	39 (25.4)	1.57 (0.53, 4.66)
Household wealth index						
Low HH wealth status(ref)	568 (16.0)	2989 (84.0)	1 (1,1)	18 (35.2)	35 (64.8)	1 (1,1)
High HH wealth status	238 (23.6)	767 (76.4)	1.19 (0.89, 1.60)	135 (76.8)	41 (23.2)	**2.29 (1.12, 4.70) ****
Exposure to mass media						
No exposure (ref)	557 (15.6)	3005 (84.4)	1 (1,1)	61 (56.4)	47 (43.6)	1 (1,1)
Exposed to either radio or TV	165 (23.9)	528 (76.1)	**1.37 (1.02, 1.84) ****	55 (75.4)	18 (24.6)	**2.02 (1.05, 3.90) ****
Exposed to both radio and TV	84 (27.5)	223 (72.5)	1.27 (0.81, 2.00)	37 (79.3)	11 (20.7)	1.47 (0.61, 3.54)
Number of under 5 children						
No child(ref)	27 (19.2)	113 (80.8)	1 (1,1)	8 (63.4)	5 (36.6)	1 (1,1)
One child	349 (17.3)	1663 (82.7)	0.95 (0.50, 1.81)	87 (69.4)	39 (30.6)	1.92 (0.48, 7.66)
Two or more	430 (17.8)	1980 (82.2)	1.10 (0.57, 2.14)	58 (64.5)	32 (35.5)	2.76 (0.66, 11.56)
Currently breastfeeding						
No(ref)	253 (15.0)	1427 (85.0)	1 (1,1)	54 (57.1)	42 (42.9)	1 (1,1)
Yes	553 (19.2)	2329 (80.8)	**1.45 (1.15, 1.84) *****	99 (74.5)	34 (25.5)	**2.02 (1.15, 3.53) ****
^b^ Perceived distance to a health facility to get medical help						
Not- big problem(ref)	367 (23.1)	1221 (76.9)	1 (1,1)	126 (71.0)	52 (29.0)	1 (1,1)
Big problem	439 (14.8)	2535 (85.2)	**0.72 (0.57, 0.91) *****	28 (54.1)	24 (45.9)	0.58 (0.32, 1.08)
Anemia status						
Non-anemic(ref)	593 (19.5)	2444 (80.5)	1 (1,1)	126 (69.4)	57 (30.6)	1 (1,1)
Anemic	213 (14.0)	1312 (86.0)	1.04 (0.83, 1.30)	27 (58.6)	19 (41.4)	1.47 (0.80, 2.72)
ANC Utilization						
No ANC visits(ref)	118 (5.4)	2061 (94.6)	1 (1,1)	17 (42.2)	26 (57.8)	1 (1,1)
At least one ANC visit	688 (28.9)	1695 (71.1)	**7.14 (5.42, 9.41) *****	136 (73.0)	50 (27.0)	**4.49 (2.16, 9.33) *****
Community level Factors						
Contextual Regions						
Agrarian(ref)	761 (17.9)	3482 (82.1)	1 (1,1)	116 (71.0)	48 (29.0)	1 (1,1)
Pastoralist	39 (13.0)	261 (87.0)	**0.54 (0.38, 0.78) *****	15 (39.4)	23 (60.6)	**0.20 (0.09, 0.46) *****
City dweller’s	6 (29.2)	13 (70.8)	**1.96 (1.09, 3.52) ****	22 (82.5)	5 (17.5)	0.96 (0.40, 2.31)

sig. at **sig. at 5% level (in bold); ***sig. at 1% level (bold); ^b^ Total figure may not add to 100 percent due to ‘do not know’ and ‘missing cases’. ref = reference group; PDs = Positive Deviants; N = Number of participants., AOR = Adjusted Odds ratios. Note: PD defined as illiterate women with use of institutional delivery services.

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
