# Peer review of "Uptake of Skilled Maternal Healthcare in Ethiopia: A Positive Deviance Approach"

_ijerph, 2020, doi:10.3390/ijerph17051712_

Round 1

Reviewer 1 Report

Referee report for the manuscript entitled "Uptake of Skilled Maternal Healthcare in Ethiopia: A Positive  Deviance Approach"

The manuscript is well written, in good shape and easy to read. The topic is of great importance for many developing countries, and for the implementation of public health policies.

I find the manuscript and its findings very interesting. However, I have a few comments.

First, I would like the author(s) to explain carefully in which differ the goal and the methodology of this paper with respect to others in the literature. The author(s) state that previous literature has focused on information at different levels (individual, community and policy levels) to identify the level of utilization of healthcare services in general, and specifically in this paper for skilled maternal healthcare. The methodology that is used in this manuscript, is essentially the same as in previous studies.  They look at the Positive Deviants (PD) and try to identify which characteristics would increase the likelihood of becoming a PD. This is basically doing the same as previous literature, and the results are also in the same line as previous papers. This is not a strong critic, as I find very interesting to try to identify the individual characteristics of a very relevant segment of individuals. However, I would like to read an explanation on why this is different to the other cited papers or to just look at the interaction of different variables as "urban or rural" and "adequate use" for example, or "level of education" or "employment status".

Second, author(s) could have used exactly the same methodology to find the individual characteristics of the Negative Deviants (ND), that we could define as those with very inadequate use (for instance, 0 or 1 visits) and still, having received formal education. Would the findings be consistent with those presented in the paper regarding the PD?

Third, the variables (individual levels of education, employment, employment of the partner, thinking of distance as a challenge, and others) in which the study is focused and that is compared with other studies in the literature, might all be highly correlated with the income level. However, I am surprised that the variable used for "household wealth index" is not more significant in the analysis. How would the author(s) interpret this? did they check for the correlation between wealth and the other variables to see if there is a different estimation strategy that might change results?

Fourth, I have a question regarding the effect of time in the results. The two waves of data are from years 2010-2011 and 2016. Maybe six years is not enough time to find significant changes in the relevant variables of the study. However, I find very strong the statement in section 4.1 "The cross-sectional nature of the study does not allow for any inference about causality. However, some of the main association in the study are likely unidirectional, e.g. education affects the use of maternal healthcare services and not vice versa". I think there might be a possibility that in time, with other policy or community variables in higher levels, as a greater access to media, also increase the use of health facilities, and that increases the level of education. Most likely, difficult to prove the existence of this direction in the causality, but still, not sure it can be ruled out.

Finally, I would have liked to read more on the policy implications of the findings in this manuscript. Is there a strategy that should be implemented for the PD or maybe also for the ND?

Author Response

Response to Reviewer 1 comments:

 The manuscript is well written, in good shape and easy to read. The topic is of great importance for many developing countries, and for the implantation of public health policies.

Response: We thank the reviewer for this comment.

1) I find the manuscript and its findings very interesting. However, I have a few comments. I would like the author (s) to explain carefully in which differ the goal and the methodology of this paper with respect to others in the literature. The author (s) state that previous literature has focused on information at different levels (individual, community and policy levels) to identify the level of utilization of healthcare services in general, and specifically in this paper for skilled maternal healthcare. The methodology that is used in this manuscript, is essentially the same as in previous studies. They look at the Positive Deviants (PD) and try to identify which characteristics would increase the likelihood of becoming a PD. This is basically doing the same as previous literature, and the results are also in the same line as previous papers. This is not a strong critic, as I find very interesting to try to identify the individual characteristics of a very relevant segment of individuals. However, I would like to read an explanation on why this is different to other cited papers or to just look at the interaction of different variables as "urban or rural" and "adequate use" for example, or "level of education" or "employment status".

Response: Thank you for the comment. As the reviewer indicates, several studies have explored factors influencing women’s utilization of skilled maternal services. However, most of these studies have looked at the general population of women and did not adopt a positive deviance approach, which we believe is quite relevant to explore applicable entry points for policies targeting vulnerable groups of women. Therefore, we believe that, by adopting a positive deviance approach, the current study provides new and useful findings to the field. We have made some edits in the background to highlight the type of studies available and indicate the scarcity of studies using positive deviance approach to study maternal utilization of skilled maternal services (see lines 47-68). Also, the references 21, 23 and 24 have been replaced by more relevant PD references (see lines 410-431). In addition, the use of multilevel analysis approach in the current study also represents a strength although such an approach has also been used in other studies

2) Author (s) could have used exactly the same methodology to find the individual characteristics of the Negative Deviants (ND), that we could define as those with very inadequate use (for instance, 0 or 1 visits) and still, having received formal education. Would the findings be consistent with those presented in the paper regarding the PD?

Response: As pointed out in this study in Section 2.3. (identification of Positive Deviants), our interest was to focus on positive deviants in order to assess important entry points for interventions targeting high-risk mothers, in this case mothers with no education. Indeed, it would also be possible to do analyses looking at negative deviants, but that was not our current aim. The reviewer’s comment might be relevant and we will explore the feasibility of such a strategy in future studies

3) The variables (individual levels of education, employment, employment of the partner, thinking of distance as a challenge, and others) in which the study is focused and that is compared with other studies in the literature, might all be highly correlated with the income level. However, I am surprised that the variable used for "household wealth index" is not more significant in the analysis. How would the author (s) interpret this? Did they check for the correlation between wealth and other variables to see if there is a different estimation strategy that might change results?

Response: As stated in the "Methods" section, we have checked for multi-collinearity using variance inflation factors (VIF) to rule out the variables that might be correlated to each other in sub-section 2.4.1. (see lines 118-120). With regards to the income variable, it is a good question as to why the associations were not more significant. Some potential explanations might be that the measure of income might not provide a thorough reflection of material assets in the study settings, the measure had to be categorized into two categories because of its distribution, it could also be that other socioeconomic indicators are more relevant than income per se. We do however not have other relevant indicators available.

4) I have a question regarding the effect of time in the results. The two waves of data are from years 2010-2011 and 2016. Maybe six years is not enough time to find significant changes in the relevant variables of the study. However, I find very strong the statement in section 4.1 "The cross-sectional nature of the study does not allow for any inference about causality. However, some of the main association in the study are likely unidirectional, e.g. education affects the use of maternal healthcare services and not vice versa". I think there might be a possibility that in time, also increase the use of health facilities, and that increases the level of education. Most likely, difficult to prove the existence of this direction in the causality, but still, not sure it can be ruled out.

Response: About the effect of time mentioned, we have explained in the methodology sub-section 2.1. (see lines 79-82) that the eligibility criteria were: being of reproductive ages (15 to 49 years), reporting at least one birth during the last five years preceding the actual survey, (i.e., 2006 – 2011 and 2012 – 2016); and participating in one of the two surveys from any region in the country. Since the EDHS collected birth history information for all live births in the five years preceding the data of the surveys. Therefore, this period was at least 10 years (2006 – 2016) and this may allow changes in the relevant variables of the study.

The other important aspect the reviewer raises is the concern on association between the factors and outcome variables:

Thank you for the comment. We have now rephrased the first sentence and deleted the second one (line 308):

"The cross-sectional analyses at the two time points limit inference about causality.”

Finally, I would have liked to read more on the policy implications of the findings in this manuscript. Is there a strategy that should be implemented for the PD or maybe also for the ND?

  Response

Thank you for pointing this out, we have added the following text at the bottom of the discussion section, line 291 – 299 (new text) as follows:

"Our study indicated that women in disadvantaged circumstances can still achieve good outcomes amidst a host of contextual barriers that usually predict poor health outcomes. Positive deviants in this context are shown to be women who have husbands with primary or above education, who have access to information through mass media, and who are involved at least with one decision making of the household. By knowing these characteristics, multiple government sectors (i.e. education, information and culture among others, alongside ministry of health) can identify women who fit the description and thus develop and amplify action plans encouraging more positive deviant behaviours".

----------------------------------------------------------------------------------------------------------------------

Reviewer 2 Report

Overall this is a well-written and comprehensive study. The study design and analysis also seem well thought-out and sound. I have only a few minor specific points as follows:

Page 1: the term ANC is not defined in the abstract on first use.

Page 4: I appreciate there may be some missing data points for various outcomes. However, the overall n’s reported in table and text don’t seem to quite add up. We are told there are 15,493 participants in the study, but 6720, 1188, 6619 and 965 add to 15,492?

Figure 1: The figure key is so small that the writing cannot be read. Either make the text larger or delete the key box and have the key in the figure legend.

END

Author Response

Response to Reviewer 2 comments:

 Overall this is a well-written and comprehensive study. The study design and analysis also seem well thought-out and sound. I have only a few minor specific point as follows:

Response: We thank the reviewer’ for the positive comments.

Page 1: the term ANC is not defined in the abstract on first use.

Response: Thank you for the comment. We have now, defined the abbreviation upon first appearance in the abstract section, line 20 as follows:

                                         " … antenatal care (ANC) …"

                           Also, we improved this section (see lines 26 - 29)

Page 4: I appreciate there may be some missing data points for various outcomes. However, the overall n’s reported in table and text don’t seem to quite add up. We are told there are 15,493 participants in the study, but 6720, 1188, 6619, and 965 added to 15,492?

Response: Please excuse this error. Based on your keen observation, we have revised the methodology sub-section 2.1. (see lines 78) that the" … 7584 women collected from 641 Enumeration (EA) (clusters) in 2016…"   AND the result descriptive (see lines 147), including title above the table placed (see lines 155) as follows:

 "Table 1. Background characteristics of 15,492 women with a live birth in the five-years preceding the 2011 or 2016 surveys (Ethiopian DHS)"

Figure 1: The figure key is so small that the writing cannot be read. Either make the text larger or delete the key box and have the key in the figure legend.

Response: Thank you, in order to be more clear, we have made the new figure key (urban/rural) larger than the former and see the new figure on lines 202 – 203. The older figure is cancelled which is below line 201.

------------------------------------------------------------------------------------------
